# Predictors of the Behavioral Intention to Participate in Saiga Antelope Conservation among Chinese Young Residents

Tingyu Yang [1], Elena Druică [2], Zhongyi Zhang [1], Yuxuan Hu [1], Giuseppe T. Cirella [3] and Yi Xie [1,*]

1   School of Economics and Management, Beijing Forestry University, Beijing 100083, China;
    yyt132179@bjfu.edu.cn (T.Y.); zzy_1991@bjfu.edu.cn (Z.Z.); yuxuan_hu@bjfu.edu.cn (Y.H.)
2   Centre for Applied Behavioral Economics, Department of Applied Economics and Quantitative Analysis,
    University of Bucharest, 030018 Bucharest, Romania; elena.druica@faa.unibuc.ro
3   Faculty of Economics, University of Gdansk, 81-824 Sopot, Poland; gt.cirella@ug.edu.pl
*   Correspondence: yixie@bjfu.edu.cn

**Abstract:** Promoting public participation is a practical move to strengthen wildlife conservation. This study focuses on saiga antelope (*Saiga tatarica*), an endangered species which has received international concern. Based on an extended version of the Theory of Planned Behavior and a sample of 536 Chinese residents aged 16–40 collected through an online survey, we applied Partial Least Squares Structural Equation Modeling to explore the predictors of the behavioral intention to participate in saiga antelope conservation. The results show that perceived behavioral control is the most influential predictor that contributes to the value of the behavioral intention, followed by injunctive norm, attitude to participation, knowledge of saiga antelope, experience of wildlife conservation, and attitude to saiga antelope, altogether explaining 48.4% of the variance of the behavioral intention. To promote public participation in saiga antelope conservation, strengthening science popularization and broadening the channels of participation are suggested.

**Keywords:** *Saiga tatarica*; wildlife conservation; theory of planned behavior; partial least squares structural equation modeling

## 1. Introduction

Wildlife conservation needs public support to cope with the poaching caused by illegal demand of wildlife products and inadequate governmental input on wildlife conservation affairs [1]. In this context, increasing public intention to participate in wildlife conservation through publicity and education activities has been a global consensus. This paper focuses on saiga antelope (*Saiga tatarica*), an endangered species that is being strictly protected by China, and aims to investigate individual behavioral intention to participate in its conservation and explore the determinants of the intention.

Saiga antelope is a Critically Endangered and Largely Depleted species in the IUCN Red List of Threatened Species [2], with five wild populations currently distributed in Russia, Kazakhstan, Uzbekistan, and Mongolia [3]. Globally, its population used to be numbered in millions. In the 19th and early 20th century, excessive hunting for its meat, hides and horns in Russia, of which large quantities of horns were exported to China, reduced the population to only a few thousand individuals [3]. Fortunately, the species received legal protection by the Soviet Union in 1919 and soon began to recover [2], reaching 2 million in the 1950s [4]. However, due to poaching, farming, and climate change [2,3], the population dropped again to 1.25 million in the mid-1970s [2], 1 million in the 1990s [4], and 26 thousand by the year of 2000 [5], causing the species Critically Endangered on the IUCN Red List in 2002. International conservation actions kicked in from around 2005, after which the status of the species has generally improved [3]. At present, though the number of mature individuals has been recovered to around 124 thousand, the recovery potential for the species is medium [2], since it is still threatened by habitat loss [4], poaching [6],

climate change, and artificial interruption of seasonal migration routes [7]. In 2019, the Convention on International Trade in Endangered Species of Wild Fauna and Flora at the 18th Conference of the Parties (CITES CoP18) passed the strict regulation of zero export quota for wild saiga specimens traded for commercial purposes, which indicates significant interest of global conservation community in strengthening saiga antelope conservation.

The horns of male saiga antelope are highly valued in Traditional Chinese Medicine [8], which is one of the factors that driving hunting [9]. Saiga antelope used to be widely distributed in northwest China [9], but became extinct in the mid-20th century due to overhunting, habitat reduction, and fragmentation of migratory routes by the closure of borders [10]. The saiga antelope has been listed as a first-class national protected species in China since 1988. To rebuild a saiga antelope population, China established the Gansu Endangered Animal Protection Centre in 1987, and started to reintroduce saiga antelope from other countries in 1988 [9,11], aiming to establish a free-ranging population in the wild someday [9]. At present, the population has been increased to over 170 individuals [9,11]. However, low genetic diversity, lack of enough breeding area, difficulties in disease prevention, and weak scientific research conditions restrict the progress of population recovery [11]. The existing conservation approach, dominated by the government with cooperation of scientific research institutions and very few local residents, is limited. In 2021, the White Paper on Biodiversity Conservation in China was issued, pointing out that an cooperative action system involving stronger government guidance, corporate action, and extensive public participation is taking shape; moreover, public participation in biodiversity conservation has grown and become more diversified in the past few years [12]. Public participation in wildlife conservation is increasingly significant in developing countries [13], though less prevalent compared to the developed countries [14]. Consequently, the cooperative action system for biodiversity conservation in China could be further strengthened by more extensive public participation.

Saiga antelope conservation needs public support as a beneficial supplementary input of the government. However, public support is much more common on star-species, such as the giant panda (*Ailuropoda melanoleuca*) [15] and African elephant (*Loxodonta*) [16], rather than non-star species, such as the saiga antelope. Similarly, social science research paid more attention to the star-species mentioned above, while saiga antelope conservation and public participation in it have not gathered enough concern. Even though, a few studies have looked at public perception and behavior related to saiga antelope, which is an essential factor that can influence its conservation. As for public awareness, Howe et al. (2012) [17] evaluated the effectiveness of a public awareness campaign on saiga antelope as a conservation intervention in Russia, and Wang and Jin (2019) [4] pointed out that laws should be strengthened to raise public awareness for their conservation in China. These studies cannot provide insight into the perceptual and behavioral dimensions of conservation-based behavior, which can act as another method conserving saiga antelope. How many and why people are concerned with wild animal protection will affect what methods are used for future intervention [18]. If more people are motivated to participate in activities that are conducive to saiga antelope conservation, the conservation process may be promoted.

To address this concern, this paper aims to identify young residents' intention to participate in saiga antelope conservation, using an extended theory of planned behavior (TPB) as theoretical background and a sample of 536 respondents to an online survey. Based on partial least squares structural equation modeling (PLS-SEM), we both assessed the statistical relevance of the predictors in the extended TPB and identified the most suitable determinant for practical interventions aimed to enhance public intention to participate in saiga antelope conservation. To our knowledge, this is the first exploratory study of this kind, rooted in behavioral psychology and promoting an interdisciplinary view on the topic. Moreover, the research on public intention to conserve saiga antelope could provide ground for understanding similarities and differences of public intention to conserve star and non-star species.

## 2. Theoretical Background and Hypotheses

### 2.1. Application of TPB in Public Intention to Participate in Wildlife Conservation

TPB was proposed by Ajzen in 1985 based on the theory of reasoned action (TRA) [19], and has been used to understand a variety of different wildlife-related behaviors for more than two decades [20]. Classical TPB believes that behavioral intention is the most direct factor affecting behavior, which is influenced by attitudes, subjective norm, and perceived behavioral control [21]. As a direct predictor for actual behavior, the stronger the intention to engage in a behavior, the more likely that behavior is [21,22].

Although research of public participation in the conservation of saiga antelope is scarce, some studies have focused on individual behavioral intention to participate in conservation of other species as well as their influencing factors based on the classical TPB framework. Aipanjiguly et al. (2003) [23] surveyed the behavioral intentions of boaters towards the conservation of manatees (*Trichechus manatus latirostris*) and found that knowledge, attitudes, and subjective norm correlated with support for manatee conservation. Lo et al. (2012) [24] studied Chinese college students' intention to support the conservation of Asian turtles (*Heosemys grandis*), and found subjective norms, attitudes toward turtle protection, and perceived behavioral control are the main predictors. Perry-Hill et al. (2014) [25] assessed the behavioral intentions towards hellbender salamander (*Cryptobranchus alleganiensis*) of landowners in Missouri and found that attitudes towards the species was a stronger and more consistent predictor of behavioral intention than basic wildlife beliefs.

The classical TPB framework has been extended by involving non-psychological factors, such as experience, age, and other demographic factors, to enrich the model [26]. Huang (2017) [27] found that people's intention to protect elephant was low in China, but people with higher degree of education and better economic status had higher intention to protect the species. Lo et al. (2012) [24] revealed that strong ethics and socio-economic variables had some statistical significant impacts on the variance of behavioral intention to support conservation of Asian turtle (*Heosemys grandis*) in China.

### 2.2. Theoretical Framework and Hypotheses Development

Since TPB is open to the inclusion of additional predictors [28], we propose an expanded approach (Figure 1) pinpointing key factors of experience and knowledge, based on the classical TPB framework and recent expanded studies [24,26,27], to better present the influence of individual psychological and non-psychological factors on his behavioral intention of saiga antelope conservation.

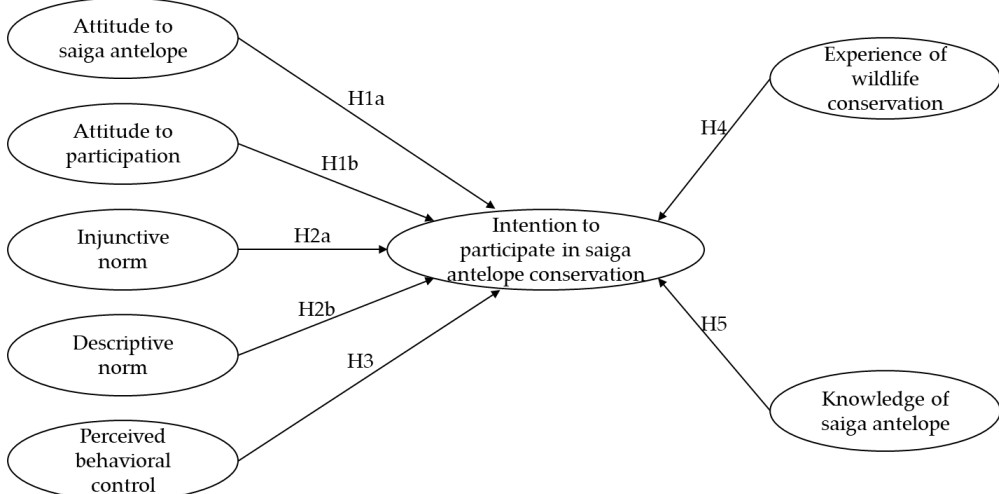

**Figure 1.** Theoretical framework and hypotheses.

In this study, the behavioral intention of saiga antelope conservation, i.e., their behavioral intention to participate in conservation activities of saiga antelope, is an explained variable. Intentions are assumed to capture the motivational factors that influence a behavior, which are indications of how hard people are willing to try and of how much of an effort they are planning to exert, in order to perform the behavior [21]. TPB starts with an explicit definition of the behavior of interest [28]. Correspondingly, the intention of saiga antelope conservation refers to the extent to which one is willing to participate in conservation activities of the species—three key categories were studied, namely, online, offline, and donation activities. Online activities contain forwarding relevant news, science articles and public service advertisements calling for saiga antelope conservation and the boycott of illegal trade of saiga horn on the Internet, as well as online supervision and reporting of illegal activities related to saiga antelope. Offline activities contain participating in offline publicity, popularizing knowledge about saiga antelope conservation to people around, and participating in habitat conservation in the field. Donation activities refer to donations for the purpose of conserving saiga antelope.

According to the classical TPB, attitude towards behavior refers to the degree to which a person has a favorable or unfavorable evaluation of the behavior in question [21]. It has been widely proven that attitudes towards conserving behavior was an important predictor of the behavioral intention to conserve wildlife, such as one's behavioral intention to conserve manatees [23] or Asian turtles [24]. In addition, some research also found that attitudes towards the species being studied and significant predictors of the behavioral intentions to conserve them [23,29,30]. Two types of attitudes are involved in this study, namely attitude to saiga antelope and attitudes to the behavior of participation. Furthermore, the factor attitude is defined as one's evaluation of saiga antelope and its conservation activities. It is generally believed that the more positive the attitudes, the stronger the behavioral intention. Therefore, two hypotheses with respect to attitudes are set as following:

**Hypothesis 1a (H1a):** *Attitude to saiga antelope positively affect the intention to participate in saiga antelope conservation.*

**Hypothesis 1b (H1b):** *Attitude to the behavior of participation positively affect the intention to participate in saiga antelope conservation.*

Subjective norms, which consist of injunctive norms and descriptive norms, refer to the perceived social pressure to perform or not to perform the behavior [21]. Subjective norms turned out to be a key determinant of the behavioral intention to participate in activities that benefit wildlife, such as being a volunteer and donating [31]. Furthermore, the perceived pressure to conserve wildlife can not only come from important others, such as friends, family members [24,31], but also from the social environment, such as government and the society [24,32]. In this study, the injunctive norm refers to the perception of pressure that one should participate in saiga antelope conservation, and the descriptive norm refers to the perception of pressure from others' participation in saiga antelope conservation. Generally, the stronger the perceived social pressure, the stronger is the behavioral intention. Therefore, two hypotheses with respect to subjective norms are set as following:

**Hypothesis 2a (H2a):** *Injunctive norm positively affect the intention to participate in saiga antelope conservation.*

**Hypothesis 2b (H2b):** *Descriptive norm positively affect the intention to participate in saiga antelope conservation.*

TPB differs from TRA by the additional predictor perceived behavioral control, which refers to people's perception of the ease or difficulty of performing the behavior [21]. Several studies identified the direct positive effect of perceived behavioral control on behavioral

intention of wildlife conservation [24,31]. In this study, perceived behavioral control refers to the perceived difficulty of participating in saiga antelope conservation. In general, the lower the perceived difficulty, the stronger the behavioral intention.

**Hypothesis 3 (H3):** *Perceived behavioral control positively affects the intention to participate in saiga antelope conservation.*

The frequency with which a behavior has been performed in the past was found to account for variance in later behavior independent of intentions [33]. Previous experience was found directly related to behavioral intention and could explain 7.2% of the variance in behavioral intention [34]. Regarding wildlife conservation, Kamrowski et al. (2014) [35] found past behavior is a significant predictor of intention to engage with light-glow reduction behaviors for marine turtle conservation, and Zhang et al. (2021) [32] verified past experience about wildlife conservation positively affected Chinese residents' willingness to protect the African elephant. In this study, experience refers to individual experience of participating in wildlife conservation activities in the past.

**Hypothesis 4 (H4):** *Experience of wildlife conservation positively affects the intention to participate in saiga antelope conservation.*

Pro-environmental behavior is motivated primarily by enhanced scientific understanding [23,24]. Turpie (2003) [36] found that one's knowledge of biodiversity was positively correlated with willingness to pay for biodiversity conservation. Studies on wildlife conservation also recognized knowledge as an influencing factor of intention to conserve the species [23,24,29]. In this study, the variable knowledge was defined as one's knowledge of the facts about saiga antelope, e.g., population. The higher the level of one's knowledge, the more likely is for one to realize the importance of conservation, and express a higher behavioral intention to participate in conservation activities.

**Hypothesis 5 (H5):** *Knowledge of saiga antelope positively affects the intention to participate in saiga antelope conservation.*

### 3. Materials and Methods

*3.1. Measurement*

The questionnaire is comprised of 31 questions organized in three sections. The first section includes 22 questions aimed to measure intention, attitudes, subjective norm, perceived behavioral control, and experience, using a 5-point Likert scale, ranging from 1 (completely disagree) to 5 (completely agree). The second section consisted of five questions measuring saiga antelope literacy. Each correct answer was given 1 point, while wrong answers were given 0. Then, a total score was calculated. Consequently, the knowledge about saiga antelope ranges from 0 (no correct answer) to 5 (all answers were correct). The third section consisted of four questions pertaining for gender, age, educational background, and annual income. Expert consultation and pre-surveying were conducted to preliminarily assess the reliability and validity of the questionnaire. The latent constructs involved in the analysis, along with their corresponding measurement items are available in Tables 1 and A1 (Appendix A) respectively.

**Table 1.** Measurement items of each variable.

| Variable | Measurement Items |
|---|---|
| Intention to participate (INT) | Degree of willingness to participate in online, offline and donation activities of saiga antelope conservation: I1-I3 |
| Attitude to saiga antelope (ATT-SAI) | Attitudes related to the value of saiga antelope: A1, A2 |
| Attitude to participation (ATT-PAR) | Attitudes related to participating in activities of saiga antelope conservation: A3-A7 |
| Injunctive norm (NOR-INJ) | Perception of the pressure that one should participate in saiga antelope conservation: N1-N3 |
| Descriptive norm (NOR-DES) | Perception of others' participation in the conservation of saiga antelope: N4-N6 |
| Perceived behavioral control (PBC) | Perception of the difficulty of participating in online, offline and donation activities of saiga antelope conservation: C1-C3 |
| Experience of wildlife conservation (EXP) | Experience of participating in activities of wildlife conservation: E1-E3 |
| Knowledge of saiga antelope (KNO) | Knowledge of the facts about saiga antelope: K1-K5 |

### 3.2. Data

Data was collected in May 2020, using Tencent Questionnaire Platform (Tencent, founded in Shenzhen, China). The English version of the questionnaire is available, see File S1 in Supplementary Material. The online survey was spread out to the participants through WeChat, a widely Chinese social media. We collected 669 questionnaires from eleven cities distributed in four regions across China, and obtained a valid sample of 536 respondents, with an effective rate of 80.12%.

The Ethical Committee of the Beijing Forestry University (No. 35 Qinghua East Road, Haidian District, Beijing, 100083, China) informed us that we didn't need special approval to carry this study. The research was anonymously conducted and there was no concern regarding the participants' privacy.

To avert the sample representativity issue, a common shortcoming of online surveys [37], we accounted for the spatial heterogeneity of the public's attitude towards wildlife conservation [38] and adopted a systematic sampling method covering first-tier, second-tier, and third-tier cities as our study areas. We included logical verification questions to test for the quality of the data [39], and eliminate the invalid data whose answer time is less than three minutes or the repetition rate of the same options is higher than 70%. The words "there are no right or wrong answers" and "data obtained will be kept confidential and used only for academic purposes" were emphasized at the beginning of the questionnaire to secure the respondents' anonymity and ensure that the data would reflect real opinions. A monetary reward, distributed in the form of WeChat Lucky Money, was given to those who passed the quality and validity test of the questionnaire to encourage responsible contributions.

In terms of sample size for a significance level of 0.05 and a power level of 0.990, the recommended minimum threshold was 407 if calculated using the inverse square root method and 385 if calculated using the gamma-exponential method.

### 3.3. Methods

We estimated our model using a partial least squares structural equation modeling (PLS-SEM) analysis [40]. With this estimation method we aimed to maximize the explained variance of the behavioral intention to participate in saiga antelope conservation, our endogenous latent outcome, as explained by the TPB predictors and the control variables. The PLS-SEM has become a widely used method in exploratory research, and in research that aims to inform practical interventions [41]. The broad adoption of this method comes not only from the fact that it does not impose any particular distributional assumptions on

the data [42], but also from its capability to estimate complex model with relative small sample sizes.

In what follows we will report the two parts of any PLS-SEM model, namely an outer (or measurement) model, assessing the relationships of the latent constructs with their corresponding indicator manifest variables, and an inner (or structural) model that estimates the observed relationships among the latent variables themselves. The results were obtained using WarpPLS 7.0 software (Ned Kock, Laredo, TX, USA).

## 4. Results

### 4.1. Respondents

Among the respondents, the proportions of men and women are almost equal. Nearly three quarters of the respondents are between 16 and 25 years old, while those between 26 and 40 years old account for the other quarter. About half of the respondents have a high school/junior college diploma or below, while the other have a bachelor's degree or above. At least half of the respondents' annual income is lower than the average level of 32,189 yuan in China in 2020, while the other respondents' income exceeds this level. Respondents from South China account for the largest proportion, followed by respondents from East, North and Central China. A complete sample description is available in Table 2.

**Table 2.** Socio-demographic characteristics of respondents (*n* = 536).

| Demographic Variable | Assessment | Frequency | Proportion (%) |
|---|---|---|---|
| Gender | male | 254 | 47.39 |
| | female | 282 | 52.61 |
| Age | 16–20 years old | 183 | 34.14 |
| | 21–25 years old | 209 | 38.99 |
| | 26–30 years old | 94 | 17.54 |
| | 31–40 years old | 50 | 9.33 |
| Educational background | junior high school and below | 30 | 5.60 |
| | high school/junior college | 243 | 45.34 |
| | bachelor's degree | 242 | 45.15 |
| | master's degree and above | 21 | 3.92 |
| Annual income | less than 30,000 yuan | 263 | 49.07 |
| | 30,000–70,000 yuan | 128 | 23.88 |
| | 70,000–150,000 yuan | 106 | 19.78 |
| | more than 150,000 yuan | 39 | 7.28 |
| Region | North China | 120 | 22.39 |
| | East China | 141 | 26.31 |
| | South China | 181 | 33.77 |
| | Central China | 94 | 17.53 |

### 4.2. Descriptive Statistics

Table A1 (Appendix A) summarizes the descriptive statistics of each observed variable in terms of mean values and standard deviations. Scores higher than 3 (the neutral position on a 1-5 Likert measurement) are indicative of a favorable perspective on the matters, while scores less than 3 show the opposite.

It can be known from the survey results of the behavioral intention that the respondents were willing to participate in saiga antelope conservation. Furthermore, online activities are shown as the most popular way to participate, followed by donation activities and offline activities.

The respondents held positive attitudes both towards saiga antelope and towards the participation behavior. As for attitude to saiga antelope, the respondents have a higher recognition on the ecological value of saiga antelope than social value. As for attitude to participation, the respondents agreed that participating in the conservation of saiga antelope

is important and interesting. Among the three categories of activities, offline activities are recognized as the most beneficial one, followed by donation and online activities.

In terms of injunctive norm, the respondents commonly agreed that they had the responsibility to participate in the conservation activities of saiga antelope. They also felt the social pressure to participate from the government (i.e., policies, laws and regulations) and people around them (such as family and friends) to some extent. In terms of descriptive norm, they perceived the effort of the government and the media to promote saiga antelope conservation, but didn't see actions from people around them.

The respondents' perception on how difficult the participation in the conservation of saiga antelope was contingent with the manner in which such activities are performed. The value of C2 is below 3, indicating participating in saiga antelope conservation through offline activities was difficult, while the values of C1 and C3 are above 3, representing two relative easier ways for the respondents to participate.

The respondents that demonstrated a lack of related experience and knowledge all scored values of experience of wildlife conservation below 3, indicating the respondents were rarely involved in wildlife conservation activities in the past. As for knowledge, the respondents correctly answered an average of 1.90 informative questions about saiga antelope, indicating a limited understanding of the species.

### 4.3. The Measurement (Outer) Model

Table 3 shows the reliability of measurement for each construct involved in the analysis. The composite reliability values range between 0.864 and 0.932, all values being above the recommended threshold of 0.6 [43]. The Cronbach's $\alpha$ values are higher than 0.7, indicative of high internal consistency [44], with one exception: attitudes in terms of value of the saiga antelope, with a Cronbach's alpha of 0.684. However, this value is only slightly below the recommendations; given the exploratory nature of the study, and the small number of items involved in this latent construct, we can accept values as low as 0.5. In addition, the corresponding composite reliability index is 0.864, a value that exceeds the recommended threshold of 0.6 and confirms that the latent construct is reliable and can be kept in the analysis. Table 3 also shows that the average variance extracted (AVE) for each composite variable is above 0.5 [45], the threshold recommended in the literature. The reliability of measurement is confirmed.

**Table 3.** Assessment of the measurement model.

| Variable | Composite Reliability | Cronbach's $\alpha$ | Average Variance Extracted (AVE) |
|---|---|---|---|
| Intention to participate | 0.910 | 0.852 | 0.772 |
| Attitude to saiga antelope | 0.864 | 0.684 | 0.760 |
| Attitude to participation | 0.932 | 0.909 | 0.733 |
| Injunctive norm | 0.898 | 0.830 | 0.746 |
| Descriptive norm | 0.877 | 0.787 | 0.707 |
| Perceived behavioral control | 0.879 | 0.794 | 0.709 |
| Experience of wildlife conservation | 0.897 | 0.827 | 0.743 |

Table A2 (Appendix A) brings further evidence that convergent validity holds. It shows the loadings of observed variables that contribute to the reflective measurement of the latent variables. The loadings range from a lower bound of 0.710 to an upper bound of 0.914, all above the required theoretical threshold of 0.7. In addition, all off-diagonal values are lower than the diagonal value for each block of measurement items. We, therefore, decide that convergent validity holds.

Table A3 (Appendix A) shows that the discriminant validity of the measurement holds as well. All block diagonal values presented in this table, corresponding to each latent construct, are higher in all cases than the corresponding off-diagonal values [46]. In addition, none of the off-diagonal correlations exceeds the recommended value of 0.8 [47].

### 4.4. The Structural (Inner) Model

Table 4 presents the estimated standardized path coefficients of the model along with the effect size of each predictor, which represents the variance of the behavioral intention explained by the predictor. Falk and Miller (1992) [48] suggest that the variance explained for endogenous variables ($R^2$) should be greater than 0.1. The $R^2$ for the behavioral intention to engage in saiga antelope conservation is 48.4%, with an adjusted $R^2$ of 47.6%. All VIF values are lower than 2.05, and the average block VIF (AVIF) is 1.406, a value below 5, which is the recommended threshold. The Tenenhaus goodness-of-fit [49] is 0.621, ranked as a large value. The model doesn't suffer from either Simpson's paradox or statistical suppression; in addition, no bivariate causality direction has been detected.

**Table 4.** Path coefficients and effect sizes, with *p*-values in parentheses.

| Variable | Standardized Path Coefficients (β)/ Significance | Effect Sizes |
|---|---|---|
| Attitude to saiga antelope | 0.097 * ($p = 0.011$) | 0.035 |
| Attitude to participation | 0.208 *** ($p < 0.001$) | 0.102 |
| Injunctive norm | 0.263 *** ($p < 0.001$) | 0.142 |
| Descriptive norm | 0.029 ($p = 0.253$) | 0.012 |
| Perceived behavioral control | 0.280 *** ($p < 0.001$) | 0.119 |
| Experience of wildlife conservation | 0.098 * ($p = 0.011$) | 0.032 |
| Knowledge of saiga antelope | 0.110 * ($p < 0.001$) | 0.025 |

*** *p*-value < 0.001; * *p*-value < 0.05.

#### 4.4.1. The Explanatory Variables

Both attitude to saiga antelope and attitude to participation show positive effects on the intention to participate in saiga conservation activities. As a consequence, both H1a and H1b are accepted. As for the predictive power of the two types of attitudes, the latter has a higher coefficient, showing that the positive attitude regarding participating in the process exerts a stronger influence on the behavioral intention than the mere value assigned to saiga antelope. Injunctive norm is a strong and positive predictor of the behavioral intention with a second highest coefficient, thus confirming H2a. However, descriptive norm is not statistically significant, which rejects H2b. The perceived behavioral control, capturing the perceived self-efficacy of each respondent, positively affected the behavioral intention with the highest coefficient, thus H3 is supported. Experience of wildlife conservation, i.e., previous engagement in activities of wildlife conservation, contributes to the behavioral intention moderately and significantly, therefore H4 is supported. Knowledge of saiga antelope, namely, one's learning of the status of saiga antelope, also improves the behavioral intention significantly, confirming H5.

Table 5 summarizes our findings in terms of accepted and rejected research hypotheses.

**Table 5.** Summary of hypothesis testing.

| | Hypothesis | Supported/Rejected |
|---|---|---|
| H1a | Attitude to saiga antelope positively affects the intention to participate in saiga antelope conservation. | Supported |
| H1b | Attitude to the behavior of participation positively affects the intention to participate in saiga antelope conservation. | Supported |
| H2a | Injunctive norm positively affects the intention to participate in saiga antelope conservation. | Supported |
| H2b | Descriptive norm positively affects the intention to participate in saiga antelope conservation. | Rejected |
| H3 | Perceived behavioral control positively affects the intention to participate in saiga antelope conservation. | Supported |
| H4 | Experience of wildlife conservation positively affects the intention to participate in saiga antelope conservation. | Supported |
| H5 | Knowledge of saiga antelope positively affects the intention to participate in saiga antelope conservation. | Supported |

4.4.2. The Control Variables

Given that our control variables were measured as categories, we relied on multigroup analysis to explore whether the model presented in Table 4 behaves differently across categories. We conducted five multigroup explorations: by gender, age, income, education, and region, and only found significant differences in coefficients by region.

Table A4 (Appendix A) shows the standardized path coefficients by region, and Table A5 (Appendix A) displays the absolute differences between the coefficients, along with the corresponding *p*-values. The most frequent differences have been identified in attitude to participation and injunctive norm. Attitude to participation has a significantly higher contribution in predicting the behavioral intention for the respondents from Central China than for the respondents from North, East, and South China. Likewise, injunctive norm has a significantly higher contribution in predicting the behavioral intention for the respondents from North China than for the respondents from East, South, and Central China. In addition, the effects of attitude to saiga antelope and experience of wildlife conservation on the behavioral intention are stronger for the respondents from South China than for the respondents from North China.

4.4.3. The Effect Sizes

Effect sizes above 0.02 are suitable for practical interventions, and values of 0.02, 0.15, and 0.35, respectively, represent small, medium, and large effects of the exogenous latent variable [50]. Table 4 shows that all statistically significant variables have effect sizes that exceed 0.02, which may make them good candidates for policy. Among which, injunctive norm has the highest contribution to the variance of the behavioral intention (0.142), followed by perceived behavioral control (0.119) and attitude to participation (0.102). These are the three predictors that have large effects and may be most effective in practice. Attitude to saiga antelope (0.035), experience of wildlife conservation (0.032), and knowledge of saiga antelope (0.025) have medium effects on the behavioral intention, which are also worthy of reference in practice.

**5. Discussion**

Wildlife conservation is inseparable from public support. This study focused on saiga antelope, a critically endangered species that has gained international attention, and explored the determinants of individual behavioral intention to conserve it. Previous studies have looked at public attitude and behavioral intention related to saiga antelope, but systematic research of the formation of public intention to conserve saiga antelope is still lacking, which can provide evidence for interventions to promote public participation. Hence, we surveyed public perception of saiga antelope and explored the determinants

of young Chinese residents' behavioral intention to participate in its conservation, using the TPB as the main theoretical background. We expanded the original TPB framework by including experience of wildlife conservation and knowledge of saiga antelope as additional predictors. Our model explains 48.4% of the variance in the behavioral intention to conserve saiga antelope, which is slightly less than the 49.1% explained in the intentions to donate money to an environmental organization (EO) for musk ox safaris [31], approximately equal to 48.1% identified by the study about Asian turtle conservation [24], but greater than 46.1% of intentions to participate in volunteer work that benefited wildlife [31], 42% of intentions to support panther recovery [29], and 41% of intention to participate in wildlife workshops [51]. Based on this comparison, we contend that our proposed model is effective in predicting public intention to conserve saiga antelope and provides insight into actionable determinants of public willingness to support wildlife conservation.

The results showed that most of the respondents had the intention to participate in saiga antelope conservation, especially through online and donation activities, which are easier to achieve. Our findings align with recent studies' finding that Chinese residents' willingness to protect wildlife was high. For instance, 53.36% of Chinese residents were willing to pay for African elephant conservation [16], and 78.5% of the residents in Huaying city, Sichuan, China were willing to pay an annual contribution for the Giant Panda Reintroduction Project [15]. The results are indicative of higher levels of awareness and willingness of Chinese people to involve in wildlife conservation now, compared to the beginning of 21st century [18]. This might be related to the development of wildlife conservation propaganda and the improvement of public awareness of wildlife conservation in recent years [52].

The significant effect of the perceived behavioral control on the behavioral intention to engage in wildlife conservation is already documented [24,31,53]. In this study, the perceived behavioral control ranks first in terms of standardized coefficients and is positively related with the behavioral intention. However, as Table A1 shows, the average values of its measuring items were no higher than around 3, indicating that participating in conservation activities of saiga antelope was not an easy task for the respondents. This is a key point to intervene to develop higher levels of public involvement. Broadening and making public know about the channels of participation should be an effective way to improve public behavioral intention to conserve saiga antelope.

Two dimensions of attitudes are involved, namely attitude to saiga antelope and attitude to the participation behavior. Most of the respondents held positive attitudes to both, indicating a good mass basis for carrying out conservation work. However, in Kalmykia, Russia where a public awareness campaign as a conservation intervention was conducted, 94% of the residents strongly agreed that saiga antelope should be protected [17], showing room for improvement in saiga conservation work in China. Both types of attitudes positively affect the behavioral intention, with attitude to participation having higher degree of influence ($\beta = 0.208$) than the attitude to the species as such ($\beta = 0.097$). These results align with similar studies employing TPB to explain the intention to engage in wildlife conservation, showing attitudes as important contributors to the behavioral intention [24,51,54]. However, effects of the attitudes towards target species to the corresponding intentions vary with species. Ranked by standardized coefficients, attitude to saiga antelope is the least influential variable in this study, while panther conservation perceptions are the strongest predictor of intentions to support panther recovery [29], and attitudes toward bats are the second strongest predictor of intentions to conserve bats [30]. Apart for differences in measurement items, respondents' familiarity with the target species may also be a cause; that is, compared to Chinese residents' familiarity with the saiga antelope, respondents of [29,30] are more familiar with the bat and panther, respectively, and their attitudes can play a more important role in the formation of the intentions.

As for subjective norms, more than half of the respondents perceived the pressure to engage in saiga antelope conservation from the government, media, people around them, and themselves, in line with that wildlife management and conservation is attracting

growing attention from all walks of life in Chinese society [55]. In predicting the behavioral intention, injunctive norm shows a significant positive effect. Moreover, it is the predictor which contributes most to the variance of the behavioral intention (effect size of 0.142), concurring with previous other findings that social influences play a decisive role in encouraging people to participate in conservation of Asian turtle [24] and Florida panther [29]. To the contrary, descriptive norm doesn't predict the behavioral intention significantly, most likely due to the fact that others' behavior on saiga antelope conservation lacks the salience required to turn it into a social norm.

Experience of wildlife conservation shows a positive effect on the behavioral intention, that is, respondents who were more involved in wildlife conservation activities before also had higher intention to conserve saiga antelope. This result aligns with another study showing that previous experience with wildlife conservation positively affect Chinese public willingness to involve in the African elephant conservation [32]. As for the intention to engage with light-glow reduction behaviors for marine turtle conservation, adding past experience with the matter improved the $R^2$ of the model [35]. Generally, those with more experience of wildlife conservation are more likely to participate in similar activities in the future.

There is little knowledge among the respondents regarding saiga antelope, similar to the knowledge about Asian turtles [24] and Chinese horseshoe crab (*Tachypleus tridentatus*) [56], showing room for information campaigns to improve Chinese public's awareness of non-star endangered species. As expected, knowledge of saiga antelope has a significant positive effect on the behavioral intention. This relation has also been confirmed in studies on behavioral intentions to support Florida panther recovery [29] and manatee conservation [23], but was not significant when it comes to the intention to support conservation of Asian turtles [24], showing different influences for different species. Nevertheless, education and information were proved to be effective ways to strengthen public understanding of wildlife as well as the behavioral intention to conserve them [57,58], which can be applied to enhance public knowledge of saiga antelope and promote people's participation in conservation activities.

Besides, some predictors behave differently by region. As for Central China, compared to other regions, the influence of attitude to participation on the behavioral intention is stronger, indicating that the respondents there attach much importance to the feelings and outcomes of participation when decide whether to participate. As for North China, i.e., Beijing and Hebei, injunctive norm is the primary predictor, which may be resulted by a more favorable social atmosphere for wildlife conservation there, according to Zhang et al. (2008) [18] that residents in Beijing were more supportive of wildlife conservation efforts than residents in east and south cities in China. Compared to North China, attitude to saiga antelope and experience of wildlife conservation are more influential to the behavioral intention in South China, showing that awareness of the values of the species and wildlife related experience play more important roles there.

To promote public participation in saiga antelope conservation, given that perceived behavioral control is the most influential predictor, it is necessary to broaden the channels for public participation in the conservation of saiga antelope and other wild animals, and make participation easy. Currently, the conservation center for breeding saiga antelope population faces challenging maters such as lack of breeding fields and poor scientific research resources. An urgent need to expand the breeding field and increase investment is needed [11]. To respond to the financial and material needs of the conservation center, opening fund-raising channels to absorb public assistance may be an option. Moreover, people can engage in wildlife conservation by participating in science projects about wildlife [59]. The conservation center cooperation with schools, enterprises, and wildlife conservation organizations, or recruit volunteers for scientific and field-oriented work, such as open community lectures and web-based seminars, are other alternatives. Field visits can also considered, which can promote conservation through education [60].

Given that injunctive norm and knowledge of the saiga antelope promote the behavioral intention, strengthening public awareness campaigns and the dissemination of knowledge related to saiga antelope can be effective interventions. The government needs to further improve and publicize the policies, laws, and regulations involving saiga antelope, and increase punishment on the persecution of saiga antelope to enhance the injunctive norm perceived by the public. To enhance people's sense of responsibility and the perceived social expectations, information campaigns, such as a media campaign taken in Russia which raised residents' awareness of saiga antelope conservation [17], may be an effective way to create a better social atmosphere for saiga antelope conservation in China. In view of people's limited understanding of saiga antelope presently, the dissemination of information related to saiga antelope needs to be enhanced to improve public learning of the species. To this end, the conservation center and relative wildlife conservation organizations can make full use of social networking APPs such as WeChat, Weibo, and Tiktok to publish articles and videos about saiga antelope and wildlife conservation, attracting netizens to browse and spread relevant content, so as to promote the online dissemination of such knowledge and information.

Some limitations of this study should be acknowledged. First, this study only takes young residents into consideration, but the relations between the variables and the effect sizes may be different when it comes to the elderly residents. Offline survey that can obtain a more representative sample should be conducted if possible. Second, apart from the explanatory variables we concerned, there remains other factors that may predict the behavioral intention to conserve saiga antelope and other species, such as trust [24] and management preferences [29]. Third, we ignored the possible relations between the explanatory variables. Though it performed effectively in this study, the model may be further improved when moderation or mediation effects are concerned.

## 6. Conclusions

In this study, we conducted an online survey and established a PLS-SEM model based on an extended TPB framework to identify the predictors of individual behavioral intention to participate in saiga antelope conservation. We found that the respondents held positive attitudes towards saiga antelope and the participation in its conservation, felt strong social and personal pressure in respect to saiga antelope conservation, and were willing to engage in conservation activities. Nevertheless, their experience of wildlife conservation and knowledge of saiga antelope were limited, and they thought participating in saiga conservation was difficult. Except for descriptive norm, the other six predictors all have significant positive effects on the behavioral intention. Perceived behavioral control is the most influential predictor that contributes to the value of the behavioral intention, followed by injunctive norm and attitude to participation, altogether explaining 36.3% of the variance of the behavioral intention. Knowledge of the saiga antelope, experience of wildlife conservation, and attitude towards saiga antelope are also important predictors, explaining another 9.2% of the variance of the behavioral intention. To promote public participation in saiga antelope conservation, broadening the channels for public participation in relative activities and strengthening information campaigns and wildlife education are suggested.

**Supplementary Materials:** The following is available online at https://www.mdpi.com/article/10.3390/d14050411/s1, File S1. Online questionnaire (English version).

**Author Contributions:** Conceptualization, T.Y., Z.Z., Y.H. and Y.X.; methodology, T.Y., E.D. and Y.X.; software, T.Y. and E.D.; validation, Y.X.; formal analysis, T.Y. and E.D.; investigation, T.Y., Z.Z. and Y.H.; resources, Y.X.; data curation, T.Y. and E.D.; writing—original draft preparation, T.Y., E.D. and Y.X.; writing—review and editing, T.Y., E.D., G.T.C. and Y.X.; visualization, T.Y.; supervision, E.D., G.T.C. and Y.X.; project administration, Y.X.; funding acquisition, Y.X. All authors have read and agreed to the published version of the manuscript.

**Funding:** This research was funded by Major Program of National Fund of Philosophy and Social Science of China Key Project (21ZDA090). The funders had no role in the design of the study; in the collection, analyses, or interpretation of data; in the writing of the manuscript; or in the decision to publish the results.

**Institutional Review Board Statement:** Ethical review and approval were waived for this study, due to the authors' institutions allowing scholars majored in social science to conduct human study without special approval.

**Informed Consent Statement:** Informed consent was obtained from all subjects involved in the study.

**Data Availability Statement:** The data presented in this study are available on request from the corresponding author.

**Acknowledgments:** The authors are also grateful to the beneficial comments of all anonymous reviewers.

**Conflicts of Interest:** The authors declare no conflict of interest.

## Appendix A

**Table A1.** Measurement items and descriptive statistics of them (*n* = 536).

| Latent Variable | Observed Variable | Item in the Questionnaire | Mean Value | Standard Deviation |
|---|---|---|---|---|
| Intention (INT) | I1 | You are willing to participate in the online activities of saiga antelope conservation. | 4.00 | 0.95 |
| | I2 | You are willing to participate in the offline activities of saiga antelope conservation. | 3.64 | 1.16 |
| | I3 | You are willing to participate in the donation activities of saiga antelope conservation. | 3.89 | 1.05 |
| Attitude to saiga antelope (ATT-SAI) | A1 | Saiga antelope has social value and is of great significance to education, cultural and scientific research. | 3.82 | 1.14 |
| | A2 | Saiga antelope has ecological value and plays an important role in maintaining the balance of ecosystem. | 4.28 | 0.97 |
| Attitude to participation (ATT-PAR) | A3 | Participating in the conservation of saiga antelope is important and valuable. | 4.35 | 0.92 |
| | A4 | Participating in the conservation of saiga antelope is interesting and gratifying. | 4.19 | 0.97 |
| | A5 | Participating in online conservation activities has a positive effect on the conservation of saiga antelope. | 4.18 | 0.96 |
| | A6 | Participating in offline conservation activities has a positive effect on the conservation of saiga antelope. | 4.31 | 0.89 |
| | A7 | Participating in donation conservation activities has a positive effect on the conservation of saiga antelope. | 4.24 | 0.89 |
| Injunctive norm (NOR-INJ) | N1 | You have the responsibility to participate in the conservation activities of saiga antelope. | 4.23 | 0.91 |
| | N2 | People around you (family, friends, etc.) think that you should participate in the conservation of saiga antelope. | 3.74 | 1.09 |
| | N3 | Policies, laws and regulations require that you should be involved in the conservation of saiga antelope. | 3.81 | 1.11 |
| Descriptive norm (NOR-DES) | N4 | People around you (family, friends, etc.) have participated in the conservation of saiga antelope. | 2.91 | 1.36 |
| | N5 | The media is actively promoting saiga antelope conservation. | 3.45 | 1.21 |
| | N6 | The government is actively conserving saiga antelope. | 3.68 | 1.15 |

**Table A1.** *Cont.*

| Latent Variable | Observed Variable | Item in the Questionnaire | Mean Value | Standard Deviation |
|---|---|---|---|---|
| Perceived behavioral control (PBC) | C1 | You find it not difficult to participate in online activities of saiga antelope conservation. | 3.25 | 1.23 |
| | C2 | You find it not difficult to participate in offline activities of saiga antelope conservation. | 2.75 | 1.16 |
| | C3 | You find it not difficult to participate in donation activities of saiga antelope conservation. | 3.24 | 1.19 |
| Experience of wildlife conservation (EXP) | E1 | You have participated in wildlife conservation through online activities. | 2.63 | 1.27 |
| | E2 | You have participated in wildlife conservation through offline activities. | 2.11 | 1.21 |
| | E3 | You have participated in wildlife conservation through donation activities. | 2.53 | 1.25 |
| Knowledge of saiga antelope (KNO) | K1 | Which is the China Special Sign for Wildlife Management and Utilization? | 0.75 | 0.43 |
| | K2 | How many is the present population of saiga antelope in China? | 0.27 | 0.45 |
| | K3 | Are there any wild populations of saiga antelope in China? | 0.11 | 0.32 |
| | K4 | What is the protection level of saiga antelope in China? | 0.61 | 0.49 |
| | K5 | What is the international trade control level of saiga antelope? | 0.15 | 0.36 |

Except for K1–K5, the minimum and maximum values of each item are all 1.00 and 5.00, respectively, with median values of 3.00. The minimum and maximum values of K1–K5 are all 0 and 1.00, respectively, with median values of 0.50.

**Table A2.** Loadings of the observed variables on the latent variables.

| | INT | EXP | NOR-INJ | NOR-DES | PBC | ATT-SAI | ATT-PAR |
|---|---|---|---|---|---|---|---|
| I1 | 0.866 | −0.004 | −0.021 | −0.041 | 0.045 | 0.016 | 0.062 |
| I2 | 0.859 | −0.054 | 0.087 | 0.053 | −0.050 | 0.016 | −0.107 |
| I3 | 0.911 | 0.054 | −0.062 | −0.011 | 0.004 | −0.030 | 0.042 |
| E1 | −0.095 | 0.879 | 0.035 | −0.043 | 0.003 | 0.038 | 0.066 |
| E2 | 0.057 | 0.841 | 0.042 | 0.052 | 0.005 | −0.061 | −0.052 |
| E3 | 0.041 | 0.866 | −0.076 | −0.007 | −0.008 | 0.021 | −0.016 |
| N1 | 0.037 | 0.058 | 0.865 | −0.111 | −0.041 | −0.023 | 0.178 |
| N2 | −0.009 | 0.002 | 0.882 | 0.034 | 0.046 | 0.010 | −0.077 |
| N3 | −0.028 | −0.062 | 0.844 | 0.078 | −0.005 | 0.013 | −0.102 |
| N4 | 0.007 | 0.233 | 0.183 | 0.710 | 0.011 | 0.003 | −0.163 |
| N5 | −0.022 | −0.097 | −0.061 | 0.914 | −0.011 | −0.027 | 0.060 |
| N6 | 0.017 | −0.087 | −0.084 | 0.885 | 0.003 | 0.025 | 0.069 |
| C1 | −0.150 | 0.012 | −0.057 | −0.019 | 0.854 | 0.042 | 0.065 |
| C2 | 0.054 | −0.027 | 0.057 | 0.133 | 0.836 | −0.078 | −0.055 |
| C3 | 0.099 | 0.015 | 0.002 | −0.114 | 0.835 | 0.035 | −0.012 |
| A1 | 0.106 | −0.008 | 0.046 | 0.015 | −0.062 | 0.872 | −0.275 |
| A2 | −0.106 | 0.008 | −0.046 | −0.015 | 0.062 | 0.872 | 0.275 |
| A3 | 0.035 | −0.022 | −0.028 | −0.043 | −0.019 | 0.272 | 0.831 |
| A4 | −0.043 | 0.003 | −0.003 | 0.048 | −0.001 | −0.028 | 0.882 |
| A5 | 0.012 | −0.002 | −0.010 | 0.018 | 0.046 | −0.019 | 0.895 |
| A6 | −0.026 | −0.002 | −0.039 | 0.012 | −0.015 | −0.069 | 0.850 |
| A7 | 0.025 | 0.024 | 0.082 | −0.040 | −0.014 | −0.153 | 0.821 |

**Table A3.** Discriminant validity of the latent variables.

|  | INT | EXPER | NOR-INJ | NOR-DES | PBC | ATT-SAI | ATT-PAR |
|---|---|---|---|---|---|---|---|
| INT | 0.879 | 0.318 | 0.534 | 0.389 | 0.410 | 0.335 | 0.482 |
| EXP | 0.318 | 0.862 | 0.255 | 0.309 | 0.339 | 0.149 | 0.161 |
| NOR-INJ | 0.534 | 0.255 | 0.864 | 0.549 | 0.229 | 0.326 | 0.503 |
| NOR-DES | 0.389 | 0.309 | 0.549 | 0.841 | 0.363 | 0.191 | 0.252 |
| PBC | 0.410 | 0.339 | 0.229 | 0.363 | 0.842 | 0.064 | 0.141 |
| ATT-SAI | 0.335 | 0.149 | 0.326 | 0.191 | 0.064 | 0.872 | 0.615 |
| ATT-PAR | 0.482 | 0.161 | 0.503 | 0.252 | 0.141 | 0.615 | 0.856 |

**Table A4.** Absolute differences in path coefficients by region.

| Variable | North China ($n = 120$) | East China ($n = 141$) | South China ($n = 181$) | Central China ($n = 94$) |
|---|---|---|---|---|
| ATT-SAI | −0.038 | 0.129 | 0.211 | 0.042 |
| ATT-PAR | 0.175 | 0.170 | 0.125 | 0.425 |
| NOR-INJ | 0.492 | 0.136 | 0.241 | 0.228 |
| NOR-DES | 0.107 | 0.154 | 0.032 | 0.082 |
| PBC | 0.215 | 0.364 | 0.284 | 0.178 |
| EXP | −0.009 | 0.078 | 0.186 | 0.128 |
| KNO | 0.127 | 0.163 | 0.149 | 0.110 |

**Table A5.** Absolute differences in path coefficients by region.

| Variable | North vs. East China | North vs. South China | North vs. Central China | East vs. South China | East vs. Central China | South vs. Central China |
|---|---|---|---|---|---|---|
| ATT-SAI | 0.167 −0.385 | 0.249 * ($p = 0.015$) | 0.081 ($p = 0.276$) | 0.082 ($p = 0.225$) | 0.086 ($p = 0.254$) | 0.168 ($p = 0.088$) |
| ATT-PAR | 0.005 ($p = 0.485$) | 0.05 ($p = 0.331$) | 0.250 * ($p = 0.024$) | 0.045 ($p = 0.339$) | 0.255 * ($p = 0.018$) | 0.300 ** ($p = 0.005$) |
| NOR-INJ | 0.356 *** ($p < 0.001$) | 0.251 ** ($p = 0.010$) | 0.264 * ($p = 0.018$) | 0.105 ($p = 0.166$) | 0.092 ($p = 234$) | 0.013 ($p = 0.457$) |
| NOR-DES | 0.047 ($p = 0.348$) | 0.075 ($p = 0.258$) | 0.025 ($p = 0.427$) | 0.122 ($p = 0.133$) | 0.072 ($p = 0.289$) | 0.05 ($p = 0.343$) |
| PBC | 0.149 ($p = 0.100$) | 0.069 ($p = 0.268$) | 0.037 ($p = 0.388$) | 0.08 ($p = 0.222$) | 0.186 ($p = 0.068$) | 0.106 ($p = 0.190$) |
| EXP | 0.086 ($p = 0.241$) | 0.194 * ($p = 0.047$) | 0.137 ($p = 0.155$) | 0.108 ($p = 0.162$) | 0.051 ($p = 0.348$) | 0.057 ($p = 0.320$) |
| KNO | 0.035 ($p = 0.384$) | 0.021 ($p = 0.426$) | 0.017 ($p = 0.499$) | 0.014 ($p = 0.449$) | 0.053 ($p = 0.341$) | 0.039 ($p = 0.377$) |

*** $p$-value < 0.001; ** $p$-value < 0.01; * $p$-value < 0.05.

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
