# Peer review of "Predictors of the Behavioral Intention to Participate in Saiga Antelope Conservation among Chinese Young Residents"

_diversity, doi:10.3390/d14050411_

Round 1

Reviewer 1 Report

Thank you for an interesting manuscript which presents an online survey of behavioral intention to participate in Saiga antelope conservation by 536 young Chinese persons from five different locations (Beijing, Hebei, Jiangsu, Guangdong and Hunan). However, the introduction does not clarly indicate why the study was made among Chinese respondents nor give a justification why the survey was implemented in a country where the antelope no longer exist in the wild. Only a few individual saigas are still kept in selected breeding centers in China, but the species cannot be found in nature there.

The authors could explain in more detail the current status of the Saiga antelope in China, reasons for extinction in the country, including its importance in illegal wildlife trade (different horn products, etc.) and actions taken by the government to support Saiga conservation locally, nationally and continentally. This would help increase the novelty and value of the study.

The authors use the concept "cooperative conservation" without defining it clearly or explaining how the approach is performing in China and in the other countries where the Saiga antelope still is found in the wild. Additionally, the interpretation of the "participation in conservation activities" in the study remains very narrow and limited. The framework of conservation activities in the online survey is limited to respondents' a) online activities, b) offline publicity and knowledge promotion, and c) donation activities. As a result, many important human dimensions of wildlife conservation are left out from the analysis. The study, for example, leaves out critical questions, such as how and where the respondents want saiga antelope to be managed, and how they are affected by the saiga antelope management decisions. It would be interesting to know which concrete conservation activities are given in the full questionnaire as measurement items to the respondent. The offline conservation activities seem to be very generally and superficially described in the manuscript, so perhaps more concrete measures, such as consumer choice decisions to avoid wildlife products, or participation in habitat management and rehabilitation, could result in somewhat different mean values.

According to the results, responses to the second section of the questionnaire that measures saiga antelope literacy (an average of 1.9 questions out of 5 correctly answered) do not grade so high (only 0.27 on the knowledge of present population, and 0.11 on the wild populations of Saiga in China), meaning that most of the 536 participants in the survey do not know the species very well. Perhaps the same results on predictors of the behavioral intention to participate in conservation could have been received if the studied species had been some other non-flagship species than saiga antelope.

The five hypotheses tested are not as such original because many previous studies have tested predictors to behavioural intention to participate in nature protection or conservation (see for example, Empidi & Emang, 2021 in Sustainability  https://doi.org/10.3390/su13084399).

Table 2 presents the demographic variables of the studied sample. It shows that over 90% of the respondents are under 30 years of age and hold at least a high school degree. Therefore the sample seems to be very homogenic, which explains why the control variables (demographic) did not identify any significant differences. The scientific soundness of the manuscript is reduced because of these above-mentioned elements and gaps.

It would have been interesting to read if the authors found any difference between the respondents from different location, e.g. between respondents from Beijing and Hunan.

In general the text is easy to read, but I recommend a language revision to be done before submitting a revised version to the journal.

I recommend a major revision of the manuscript.

Author Response

COMMENT 1

Thank you for an interesting manuscript which presents an online survey of behavioral intention to participate in Saiga antelope conservation by 536 young Chinese persons from five different locations (Beijing, Hebei, Jiangsu, Guangdong and Hunan). However, the introduction does not clearly indicate why the study was made among Chinese respondents nor give a justification why the survey was implemented in a country where the antelope no longer exist in the wild. Only a few individual saigas are still kept in selected breeding centers in China, but the species cannot be found in nature there.

The authors could explain in more detail the current status of the Saiga antelope in China, reasons for extinction in the country, including its importance in illegal wildlife trade (different horn products, etc.) and actions taken by the government to support Saiga conservation locally, nationally and continentally. This would help increase the novelty and value of the study.

  • RESPONSE 1

We are grateful to your beneficial comments. According to the above suggestions, we added detailed description of the extinction reasons of saiga antelope and its status quo in China, as well as the actions taken by the government of China to conserve the species. See the revised content in Introduction for details (line 54-69).

COMMENT 2

The authors use the concept "cooperative conservation" without defining it clearly or explaining how the approach is performing in China and in the other countries where the Saiga antelope still is found in the wild. Additionally, the interpretation of the "participation in conservation activities" in the study remains very narrow and limited. The framework of conservation activities in the online survey is limited to respondents' a) online activities, b) offline publicity and knowledge promotion, and c) donation activities. As a result, many important human dimensions of wildlife conservation are left out from the analysis. The study, for example, leaves out critical questions, such as how and where the respondents want saiga antelope to be managed, and how they are affected by the saiga antelope management decisions. It would be interesting to know which concrete conservation activities are given in the full questionnaire as measurement items to the respondent. The offline conservation activities seem to be very generally and superficially described in the manuscript, so perhaps more concrete measures, such as consumer choice decisions to avoid wildlife products, or participation in habitat management and rehabilitation, could result in somewhat different mean values.

  • RESPONSE 2

Agreed. The “cooperative conservation” has been defined more clearly.

In the questionnaire, the three types of saiga antelope conservation activities were introduced to the respondents as follows.

Online activities contain forwarding relevant news, science articles and public service advertisements calling for saiga antelope conservation and the boycott of illegal trade of saiga horn on the Internet, as well as online supervision and reporting of illegal activities related to saiga antelope. Offline activities contain participating in offline publicity, popularizing knowledge about saiga antelope conservation to people around, and participating in habitat conservation in the field. Donation activities refer to donations for the purpose of conserving saiga antelope. We have revised the description of this part in the text (line 143-152).

COMMENT 3

According to the results, responses to the second section of the questionnaire that measures saiga antelope literacy (an average of 1.9 questions out of 5 correctly answered) do not grade so high (only 0.27 on the knowledge of present population, and 0.11 on the wild populations of Saiga in China), meaning that most of the 536 participants in the survey do not know the species very well. Perhaps the same results on predictors of the behavioral intention to participate in conservation could have been received if the studied species had been some other non-flagship species than saiga antelope.

  • RESPONSE 3

We have no intention to say this comment is not reasonable. However, according to previous studies that took knowledge of the studied species into consideration, the knowledges didn’t always affect the intentions to conserve the species significantly. Moreover, it is possible that people’s knowledge of a non-flagship is too undifferentiated to affect the intention significantly. Therefore, without the model estimation, we were not sure whether knowledge of saiga antelope could predict the intention we wanted to explain significantly. Only through the empirical analysis can we confirm this. At last, in this study, we verified that even for saiga antelope, a non-flagship species that respondents were commonly not much familiar with, their knowledge of it could be differentiated and indeed played a role in the formation of the intention to conserve it.

COMMENT 4

The five hypotheses tested are not as such original because many previous studies have tested predictors to behavioural intention to participate in nature protection or conservation.

  • RESPONSE 4

Disagreed. The hypotheses are proposed based on Ajzen’s TPB and some other previous studies, which are mentioned in Theoretical Background and Hypotheses. TPB is already a well-developed and widely used theory that can effectively predict a variety of intentions, and is also applied in the article you list as an example, so the hypotheses are of course not original. Moreover, the effectiveness and applicability of the theory which have been confirmed in other studies of nature conservation are exactly why it is chosen to be the theoretical basis of this study. We would like to emphasize that the aim to test the hypotheses in this study is not to test the theory, but to find the reliable predictors of the behavioral intention. Therefore, despite the non-originality, the hypotheses need to be tested.

COMMENT 5

Table 2 presents the demographic variables of the studied sample. It shows that over 90% of the respondents are under 30 years of age and hold at least a high school degree. Therefore the sample seems to be very homogenic, which explains why the control variables (demographic) did not identify any significant differences. The scientific soundness of the manuscript is reduced because of these above-mentioned elements and gaps.

  • RESPONSE 5

Agreed. It is true that the demographic characteristics of the sample are not various enough, which is one of the limitations of this study. Therefore, we put the demographic variables in the model only as control variables and focus on the influence of attitudes, norms, PBC, experience and knowledge on the intention. Compared to not concerning the demographic variables, regarding them as control variables may improve the stability of the results of the explanatory variables.

COOMENT 6

It would have been interesting to read if the authors found any difference between the respondents from different location, e.g. between respondents from Beijing and Hunan.

  • RESPONSE 6

Agreed. In order to explore the differences between the respondents from different locations, we conducted a multigroup analysis by region and found some predictors behaved differently between different regions. The results have been added and discussed, see part 4.4.2 and part 5.

COMMENT 7

In general the text is easy to read, but I recommend a language revision to be done before submitting a revised version to the journal.

  • RESPONSE 7

Agreed. Our co-author, a native English speaker, has read through the new MS again. We will have the pleasure to make further language correction if it is needed.

Reviewer 2 Report

Manuscript ID: diversity-1648153

Predictors of the Behavioral Intention to Participate in Saiga Antelope (Saiga tatarica) Conservation Among Chinese Young Residents

by Tingyu Yang, Elena Druică, Zhongyi Zhang, Yuxuan Hu, Giuseppe T. Cirella, Yi Xie

Review

Saiga protection is very important internationally, not only in China. Therefore, in the very beginning, global and green status of the species must be mentioned, CR globally:

IUCN SSC Antelope Specialist Group. 2018. Saiga tatarica. The IUCN Red List of Threatened Species 2018: e.T19832A50194357. . Accessed on 19 March 2022.

And in the Green list – LD, largely depleted:

Milner-Gulland, E.J. 

Reader should be informed about the species distribution, and especially about species presence/absence in China. Species distribution map, as Figure 1, is highly recommended. I wonder, if it is expressed contradiction between data of IUCN, saying that species is extinct in China, and species strictly protected status in the country? This requires explanation in the introduction.

My second general remark concerns structure of the manuscript: section 2 should be incorporated into 3, Material and Methods. Part of the mentioned text, e.g., Lines 89 to 116, should go to discussion.

I have also several other comments, listed below. But still, manuscript deserves publication after revision.

Title

Title is long already, I would recommend remove Latin name of the species from title, and vice versa – use only Latin name in Keywords, to avoid full duplicate

Abstract and keywords

Please exclude all abbreviations from abstract and keywords, explaining these in the text under first mentioning.

Line 23: strengthening and broadening?

Introduction

As said above, Introduction requires species distribution map and broader explanation of species status, including 2 recommended references.

Line 57: species name not full, add

Line 58: mistype

Lines 82 to 87 – exclude

Material and methods

Part of the former chapter 2 should be incorporated here, hypotheses being central part of the chapter.

Line 220 to 222: use past tense, e.g., informed

Line 232: was reward offered initially, or given afterwards?

Line 252: software should be referenced

Results

Please avoid repetitions, such as Line 255 – number of respondents already given above.

Text in Lines 255 to 262 should not repeat values, later given in the Table 2.

Current 4.2 chapter is clumsy, wrong title and then 6 small subchapters, where presented values repeat those given in the Table A1. Advice is to make chapter with general title and not split it into the small parts, presenting broader explaining of what is shown in the Table A1.

Line 345: was effect size mentioned in Material and Methods?

Conclusions

This part of the manuscript is definitely too long.

Most of this text, not only Line 477 to 483 should go to Discussion, maybe to the very beginning of it.

Some other parts of the Conclusions are, in fact, also Discussion.

Please try to give maximum two short paragraphs, showing your main findings in the condensed way.

Back matter

Please note, if there was any role of the funders in the design of the study; in the collection, analyses or interpretation of data; in the writing of the manuscript, or in the decision to publish the results must be declared in this section. If there is no role, please state “The funders had no role in the design of the study; in the collection, analyses, or interpretation of data; in the writing of the manuscript, or in the decision to publish the results”

References

Please stick to the journal Template:

  1. Journal names must be abbreviated (where applicable)
  2. Rage range should be separated by a long dash
  3. Mistypes in DOI numbers, Lines 567, 502
  4. Lines 568 and 571 – is this the same journal?

Author Response

COMMENT 1

Saiga protection is very important internationally, not only in China. Therefore, in the very beginning, global and green status of the species must be mentioned, CR globally: IUCN SSC Antelope Specialist Group. 2018. Saiga tatarica. The IUCN Red List of Threatened Species 2018: e.T19832A50194357. https://dx.doi.org/10.2305/IUCN.UK.2018-2.RLTS.T19832A50194357.en. Accessed on 19 March 2022. And in the Green list – LD, largely depleted: Milner-Gulland, E.J. 2021. Saiga tatarica (Green Status assessment). The IUCN Red List of Threatened Species 2021: e.T19832A1983220213.Accessed on 19 March 2022.

  • RESPONSE 1

We are grateful to your beneficial comments. The status of saiga antelope is added to the beginning of the second paragraph (line 32-48).

COMMENT 2

Reader should be informed about the species distribution, and especially about species presence/absence in China. Species distribution map, as Figure 1, is highly recommended. I wonder, if it is expressed contradiction between data of IUCN, saying that species is extinct in China, and species strictly protected status in the country? This requires explanation in the introduction.

  • RESPONSE 2

Agreed. We have enriched the description on the current status of saiga antelope in China (line 55-69). We note that Cui et al. (2017) have studied the distribution of saiga antelope in China and drew a distribution map of saiga antelope in China (https://www.nature.com/articles/srep44200). Therefore, we believe that we had better not repeatedly present a map in our study.

COMMENT 3

My second general remark concerns structure of the manuscript: section 2 should be incorporated into 3, Material and Methods. Part of the mentioned text, e.g., Lines 89 to 116, should go to discussion.

  • RESPONSE 3

We are grateful for this comment. However, after careful consideration and full discussion, we decided to keep the current structure of section 2 mainly because of the following two important reasons.

First, this section aims to illustrate the reason of applying TPB and why additional variables were added through giving a review on previous literature that studied human dimensions of wildlife conservation based on TPB, which provides an essential theoretical basis of the development of the hypotheses. If Lines 89 to 116 were removed from this section, the hypotheses will lack sufficient theoretical and empirical evidence. Thus, a review of relevant studies is necessary for the development of hypotheses.

Second, the structure of this manuscript is common in other papers published by MDPI that applying TPB. For instance, according to an article of Elena Druică (https://doi.org/10.3390/foods10081877), one of our co-authors, which studied the determinants of fast-food consumption using TPB, the literature review and hypotheses development are arranged in a separate section to provide information about the relevance of the TPB in predicting food choice and sets the research hypotheses. Similarly, another study on farmers’ intention towards participation in the management and conservation of wetlands (https://doi.org/10.3390/land10080860) also put the reason of using and extending TPB in a separate section. It shows that using a separate section to describe theoretical basis and hypotheses development is a common practice.

COMMENT 4

Title

Title is long already, I would recommend remove Latin name of the species from title, and vice versa – use only Latin name in Keywords, to avoid full duplicate.

  • RESPONSE 4

Agreed. The Latin name has been removed from the title and kept in the Keywords.

COMMENT 5

Abstract and keywords

Please exclude all abbreviations from abstract and keywords, explaining these in the text under first mentioning.

Line 23: strengthening and broadening?

  • RESPONSE 5

Agreed. The abbreviations in the abstract and keywords have been removed. The language of the manuscript has been revised carefully.

COMMENT 6

Introduction

As said above, Introduction requires species distribution map and broader explanation of species status, including 2 recommended references.

Line 57: species name not full, add

Line 58: mistype

Lines 82 to 87 – exclude

  • RESPONSE 6

Agreed.

The introduction has been enriched and the reason of not adding the map has been illustrated in RESPONSE 2.

We’re confused with the problem you pointed out in Line 57 since the species name is already full. Did you mean the Latin name? If you mean that, it has been mentioned in the first paragraph.

The mistype in line 58 has been corrected. Lines 82 to 87 have been excluded.

COMMENT 7

Material and methods

Part of the former chapter 2 should be incorporated here, hypotheses being central part of the chapter.

Line 220 to 222: use past tense, e.g., informed

Line 232: was reward offered initially, or given afterwards?

Line 252: software should be referenced

  • RESPONSE 7

We are grateful for these comments.

The reason of not changing the structure of section 2 has been explained in RESPONSE 3.

The tense in Lines 220-222 has been corrected.

The reward mentioned in Line 232 was given afterwards, in the process of questionnaire screening, to the respondents whose questionnaires were valid, that is, whose answer time exceeded three minutes and repetition rate of the same options were lower than 70%.

We are confused about the comment on Line 252. The software we applied is already mentioned.

COMMENT 8

Results

Please avoid repetitions, such as Line 255 – number of respondents already given above.

Text in Lines 255 to 262 should not repeat values, later given in the Table 2.

Current 4.2 chapter is clumsy, wrong title and then 6 small subchapters, where presented values repeat those given in the Table A1. Advice is to make chapter with general title and not split it into the small parts, presenting broader explaining of what is shown in the Table A1.

Line 345: was effect size mentioned in Material and Methods?

  • RESPONSE 8

Agreed.

The repeated descriptions in Lines 255 to 262 have been removed.

The description of the survey results has been reorganized in section 4.2 without being separated, and the values given in Table A1 are not repeated. Explanations of the results have been enriched.

The effect size is first mentioned in the first paragraph of section 4.4, and we added an explanation for it as a revision.

COMMENT 9

Conclusions

This part of the manuscript is definitely too long.

Most of this text, not only Line 477 to 483 should go to Discussion, maybe to the very beginning of it.

Some other parts of the Conclusions are, in fact, also Discussion.

Please try to give maximum two short paragraphs, showing your main findings in the condensed way.

  • RESPONSE 9

Agreed. We have shortened the conclusion section into one paragraph, which contains only the main findings of this study, and put the other parts into discussion section.

COMMENT 10

Back matter

Please note, if there was any role of the funders in the design of the study; in the collection, analyses or interpretation of data; in the writing of the manuscript, or in the decision to publish the results must be declared in this section. If there is no role, please state “The funders had no role in the design of the study; in the collection, analyses, or interpretation of data; in the writing of the manuscript, or in the decision to publish the results”

  • RESPONSE 10

Agreed. The statement has been added to the back matter.

COMMENT 11

References

Please stick to the journal Template:

  1. Journal names must be abbreviated (where applicable)
  2. Rage range should be separated by a long dash
  3. Mistypes in DOI numbers, Lines 567, 502
  4. Lines 568 and 571 – is this the same journal?
  • RESPONSE 11

Agreed. We have modified the references carefully according to the layout style of the journal, including the abbreviations and DOI numbers.

Round 2

Reviewer 1 Report

The quality of the revised manuscript has improved a lot.

The authors should consider removing the p-values and β from the Abstract in the same way they edited the text in chapter 4.4.1. The explanatory variables.

The introduction is now more comprehensive and justifies the study on saiga antelope in the intention to participate in conservation setting. The edits have also made the manuscript easier to read.

However, the authors do not clearly define what does "cooperative conservation approach" mean in this study. For example in 2005, G. A. Norton said in a conference in U.S.A. that the aim of cooperative conservation is '"to energize citizen-conservationists." The White House Executive Order defines the term "cooperative conservation" ...means actions that relate to use, enhancement, and enjoyment of natural resources, protection of the environment, or both, and that involve collaborative activity among Federal, State, local, and tribal governments, private for-profit and nonprofit institutions, other nongovernmental entities and individuals.

It would be good if the authors would use a published reference to define the concept "cooperative conservation" in the introduction of the manuscript.

In chapter 2, second paragraph, the authors present previous studies that have focused on individual behavioral intention to participate in conservation of other species  as well as their influencing factors based on the classical TPB framework. Here, they include a reference that does not at all focus on intention to participate in conservation but rather on intention to kill the species, namely  "Marchini and Macdonald (2012) [23] examined the role of ranchers’ perception, norms, attitudes and intention concerning jaguar (Panthera onca) killing and found that the impact of jaguars on livestock, together with fear, and personal and social motivations, were major predictors of a rancher’s intention to kill jaguars", so perhaps the authors may reconsider keeping this as a relevant reference to the paragraph content or revising the text a bit.

The discussion chapter reflects the results very well and has improved significantly. However, there is one inconsistency that may need clarifications, namely Descriptive norm (NOR-DES) items in the questionnaire are: N4 People around you (family, friends, etc.) have participated in the conservation of saiga antelope; N5 The media is actively promoting saiga antelope conservation and N6 The government is actively conserving saiga antelope. In the hypothesis testing, Descriptive norm positively affects the intention to participate in saiga antelope conservation was rejected. However, in the Discussion chapter second last paragraph (677-684), the authors recommend media campaigns to raise residents' awareness of saiga antelope conservation in China; and that the government needs to improve policies, laws and regulations involving saiga antelope. These suggestions in the paragraph are categorized under subjective norm and knowlegde of saiga, so there seems to be some sort of inconsistency between the discussion and the results, which may require further editing by the authors.

Author Response

COMMENT 1

The authors should consider removing the p-values and β from the Abstract in the same way they edited the text in chapter 4.4.1. The explanatory variables.

  • RESPONSE 1

We are grateful to the first reviewer’s beneficial comments. According to the above comment, we removed the p-values and β values from the abstract.

COMMENT 2

The introduction is now more comprehensive and justifies the study on saiga antelope in the intention to participate in conservation setting. The edits have also made the manuscript easier to read.

However, the authors do not clearly define what does "cooperative conservation approach" mean in this study. For example in 2005, G. A. Norton said in a conference in U.S.A. that the aim of cooperative conservation is "to energize citizen-conservationists." The White House Executive Order defines the term "cooperative conservation" ...means actions that relate to use, enhancement, and enjoyment of natural resources, protection of the environment, or both, and that involve collaborative activity among Federal, State, local, and tribal governments, private for-profit and nonprofit institutions, other nongovernmental entities and individuals.

It would be good if the authors would use a published reference to define the concept "cooperative conservation" in the introduction of the manuscript.

  • RESPONSE 2

We are grateful for this comment. The definition “cooperative conservation approach” has been changed into “cooperative action system”, which comes from the White Paper on Biodiversity Conservation in China issued in 2021. The cooperative action system refers to a biodiversity conservation system involving stronger government guidance, corporate action, and extensive public participation. The modified description can be found in lines 74-83.

COMMENT 3

In chapter 2, second paragraph, the authors present previous studies that have focused on individual behavioral intention to participate in conservation of other species as well as their influencing factors based on the classical TPB framework. Here, they include a reference that does not at all focus on intention to participate in conservation but rather on intention to kill the species, namely "Marchini and Macdonald (2012) [23] examined the role of ranchers’ perception, norms, attitudes and intention concerning jaguar (Panthera onca) killing and found that the impact of jaguars on livestock, together with fear, and personal and social motivations, were major predictors of a rancher’s intention to kill jaguars", so perhaps the authors may reconsider keeping this as a relevant reference to the paragraph content or revising the text a bit.

  • RESPONSE 3

Thanks for your careful review and comments. We included the study of Marchini and Macdonald (2012) mainly because of the following reason. Although the explained variable of the study is the intention to kill the species but not the intention to conserve it, the aim of Marchini and Macdonald (2012) is to understand the underlying causes of jaguar killing, which is the most urgent issue in jaguar conservation, so as to curb the jaguar killing and conserve the species. Therefore, we regarded the article as a study concerning wildlife conservation before. After reading your comments, we realized that the study of jaguar indeed doesn’t relate to our study very directly, and decided to remove it from chapter 2.

COMMENT 4

The discussion chapter reflects the results very well and has improved significantly. However, there is one inconsistency that may need clarifications, namely Descriptive norm (NOR-DES) items in the questionnaire are: N4 People around you (family, friends, etc.) have participated in the conservation of saiga antelope; N5 The media is actively promoting saiga antelope conservation and N6 The government is actively conserving saiga antelope. In the hypothesis testing, Descriptive norm positively affects the intention to participate in saiga antelope conservation was rejected. However, in the Discussion chapter second last paragraph (677-684), the authors recommend media campaigns to raise residents' awareness of saiga antelope conservation in China; and that the government needs to improve policies, laws and regulations involving saiga antelope. These suggestions in the paragraph are categorized under subjective norm and knowlegde of saiga, so there seems to be some sort of inconsistency between the discussion and the results, which may require further editing by the authors.

  • RESPONSE 4

We are grateful to the comments. I found that I miswrote the “subjective norm” at the beginning of the paragraph, which should be “injunctive norm”. I apologize for this mistake and have made the correction. Furthermore, I would like to explain about the suggestions.

Firstly, I would like to clarify the difference between descriptive norm (NOR-DES) and injunctive norm (NOR-INJ). The descriptive norm refers to one’s perception of others’ participation behaviors in saiga antelope conservation, while the injunctive norm emphasizes one’s perception of the social pressure that he should participate in saiga antelope conservation. In our study, we measured the two variables, i.e., NOR-INJ and NOR-DES, from three aspects, respectively. As you have mentioned above, NOR-DES was measured by the participation of people around (N4), the media (N5), and the government (N6). NOR-INJ was measured by the pressure to participate from oneself (N1), people around (N2), and policies, laws and regulations (N3).

Secondly, according to the results, NOR-INJ is a strong predictor, indicating that the respondents’ perception of the pressure to participate in saiga antelope conservation can enhance their intention to participate. More precisely, if one has a higher sense of the responsibility to participate, perceived higher expectations from people around that he should participate, and perceived higher pressure from policies, laws and regulations that he should participate, then his intention to participate will be higher. According to this, we suggested that “The government needs to further improve and publicize the policies, laws, and regulations… to enhance the injunctive norm”. As for the suggestion of “strengthening information campaigns and wildlife education”, we don’t mean media campaigns only, but want to stress that “the dissemination of information related to saiga antelope needs to be enhanced to improve public learning of the species”. We mentioned the media campaign from the point that it is one of the ways to spread knowledge of saiga antelope, as well as related policies, laws and regulations, rather than because the participation of media is an item that measured NOR-DES.

To make the discussion more easily to read, this part of discussion has been modified, see lines 583-597.

Reviewer 2 Report

I accept changes done in revision. The only thing should be done :

Line 932 and rest of References, page range should use long dash:

not 

Res. 2019, 32, 49-53,

but 

Res. 2019, 32, 49–53,

However, if this will be done by Editors, I have no other comments.

Author Response

COMMENT 1

Line 932 and rest of References, page range should use long dash:

not

Res. 2019, 32, 49-53,

but

Res. 2019, 32, 49–53,

  • RESPONSE 1

We are grateful to the second reviewer’s careful review and beneficial comments. The format has been corrected.